# Rapid evolution of mutation rate and spectrum in response to environmental and population-genetic challenges

Wen Wei [1,4] ✉, Wei-Chin Ho [1,4], Megan G. Behringer[1,2,3], Samuel F. Miller[1], George Bcharah[1] & Michael Lynch [1] ✉

Ecological and demographic factors can significantly shape the evolution of microbial populations both directly and indirectly, as when changes in the effective population size affect the efficiency of natural selection on the mutation rate. However, it remains unclear how rapidly the mutation-rate responds evolutionarily to the entanglement of ecological and population-genetic factors over time. Here, we directly assess the mutation rate and spectrum of *Escherichia coli* clones isolated from populations evolving in response to 1000 days of different transfer volumes and resource-replenishment intervals. The evolution of mutation rates proceeded rapidly in response to demographic and/or environmental changes, with substantial bidirectional shifts observed as early as 59 generations. These results highlight the remarkable rapidity by which mutation rates are shaped in asexual lineages in response to environmental and population-genetic forces, and are broadly consistent with the drift-barrier hypothesis for the evolution of mutation rates, while also highlighting situations in which mutator genotypes may be promoted by positive selection.

As organisms frequently encounter environmental changes, and mutation is the ultimate source of evolutionary material, studying the response of the mutation rate to population settings helps us better understand the fate and dynamics of populations in novel environments. Mutation rates are plastic and evolvable. For example, mutation rates of bacteria have been found to be plastically elevated in several different stressful environments, such as extreme starvation, exposure to antibiotics, and high population densities[1–8]. Furthermore, hyper-mutator alleles, or mutations that increase spontaneous mutation rates, can reach fixation in microbial populations evolved under stressful environments. For example, in glucose-limited media, *Escherichia coli* can evolve a higher mutation rate within ~2400 generations[9–11].

Different theories have been proposed to explain the evolution of elevated mutation rates in stressful environments. One idea is that hypermutator alleles can hitchhike with beneficial mutations and reach fixation within a population adapting to a novel environment[12–15]. Alternatively, the evolution of elevated mutation rates might result from shrinkage of the effective population size ($N_e$) under a stressful environment[16,17]. Lower mutation rates are expected to be commonly favoured by natural selection due to the prevalence of deleterious mutations. As the power of natural selection to reduce mutation rates is constrained by the magnitude of genetic drift defined by $N_e$, populations with reduced $N_e$ are expected to be unable to resist the accumulation of mutations mildly deleterious to replication fidelity[18,19].

Although prior studies on mutation-rate evolution tend to focus on homogeneous environments, natural environments tend to be spatially and temporally heterogeneous, as with fluctuating supplies of nutrients. A few theoretical studies considering heterogeneous environments suggest that the level of heterogeneity can affect mutation-

[1]Biodesign Center for Mechanisms of Evolution, Arizona State University, Tempe, AZ 85287, USA. [2]Department of Biological Sciences, Vanderbilt University, Nashville, TN 37232, USA. [3]Department of Pathology, Microbiology, and Immunology, Vanderbilt University Medical Center, Nashville, TN 37232, USA. [4]These authors contributed equally: Wen Wei, Wei-Chin Ho. ✉e-mail: wwei39@asu.edu; mlynch11@asu.edu

rate evolution. For example, hypermutators are expected in theory to be the most prevalent in environments where conditions fluctuate at intermediate frequencies, leading to a non-monotonic relationship between mutation-rate evolution and environmental variation[20,21]. However, empirical studies on mutation-rate evolution in complex environments are lacking.

In this work, we show that the evolution of mutation rates proceeded rapidly in response to demographic and/or environmental changes. To address how much the genetic adaptation to fluctuating environments impacts mutation-rate evolution, we used experimentally evolved *E. coli* lines as a model system to understand the ecological effects of fluctuating resource availability on the dynamics of mutation-rate evolution. Specifically, the experimental evolution of *E. coli* was performed in rich media with three different resource-replenishing cycles (Fig. 1): L1 (transferred daily with 1/10 dilution), L10 (transferred every 10 days with 1/10 dilution), and L100 (transferred every 100 days with 1/10 dilution). The environmental stress in L100 is presumably the strongest, which is evident by the low observed carrying capacity relative to L10 and L1[22]. To further compare the experimental results in scenarios with a reduced $N_e$ by stronger bottlenecks, two additional schemes of experimental evolution were studied: M1 (1/10⁴ dilution, transferred daily) and S1 (1/10⁷ dilution, transferred daily). The schemes L1, M1, and S1 allow comparison across different $N_e$ (respective $N_e \sim 2.5 \times 10^9$, $7.5 \times 10^6$, and $1.3 \times 10^3$). For each of the five transfer schemes, to further determine how initial mutational features affect subsequent mutation-rate evolution, within each transfer scheme, two different starting genetic backgrounds were examined: a wild-type (WT) strain, and a strain with an inactivated mismatch-repair (MMR-) pathway. With two replicate populations for each genetic background and transfer-scheme combination ($2 \times 2 \times 5 = 20$), and two additional WT L10 populations (Supplementary Fig. 1), we studied a total of 22 experimental populations. After 1000 days of evolution, we isolated two clones from each of the 22 populations, as well as two independent replicates from the ancestral WT and MMR- progenitor clones. To estimate the mutation rates of these clonal isolates, we perform a mutation-accumulation (MA) assay

on each of them, followed by whole-genome sequencing (MA/WGS) (Fig. 1; Supplementary Data 1–4), a procedure that yields essentially unbiased mutation-rate estimates by capturing mutations in an effectively neutral manner[23].

## Results

### Evolution of high mutation rates in response to intermediate resource-replenishment cycles

The mean single-nucleotide mutation (SNM) rates in ancestral WT and MMR- progenitors are respectively $3.5 \times 10^{-10}$ and $2.4 \times 10^{-8}$ per site per generation (Fig. 2a, b). In addition, the mean small-indel mutation (SIM, ≤4 bp) rates in ancestral WT and MMR- progenitors are respectively $6.3 \times 10^{-11}$ and $5.0 \times 10^{-9}$ per site per generation (Fig. 2c, d). These measurements are similar to those reported in the previous literature[24].

In comparing treatments with different lengths of resource-replenishment cycles (L1, L10, and L100), we found that the most extreme evolutionary changes in mutation rates were found in clones isolated from populations cultivated in L10. For the WT background, seven out of eight clones show apparently much higher mutation rates than the ancestor WT clones. Overall, the mean SNM and SIM rate significantly increased to be 121.4-fold ($P = 4.4 \times 10^{-44}$) and 77.3-fold ($P = 2.5 \times 10^{-47}$) of the WT ancestral value, respectively. The mean structural variation mutation (SVM, > 4 bp variants) rate for L10 WT clones is significantly increased by 1.7-fold ($P = 4.3 \times 10^{-5}$) of the ancestral value. Note that the clone L10-D1 shows a very different mutation rate compared to the other clone from the same population, L10-D2. Such intra-populational variation of mutation rates can arise from the recurrent input of mutation-rate variants during clonal evolution. For the MMR- background, clones from the L10 populations also generally show significant increases in SNM rates (by ~4.4-fold; $P = 1.5 \times 10^{-10}$) and SVM rates (by ~2.2-fold, $P = 2.4 \times 10^{-3}$) relative to the ancestral value (Fig. 2; Supplementary Fig. 2), even though the significance probably mostly comes from the clones of the population L10-A, as the mutation rates in the clones of the population L10-B and the ancestral MMR- clones are similar.

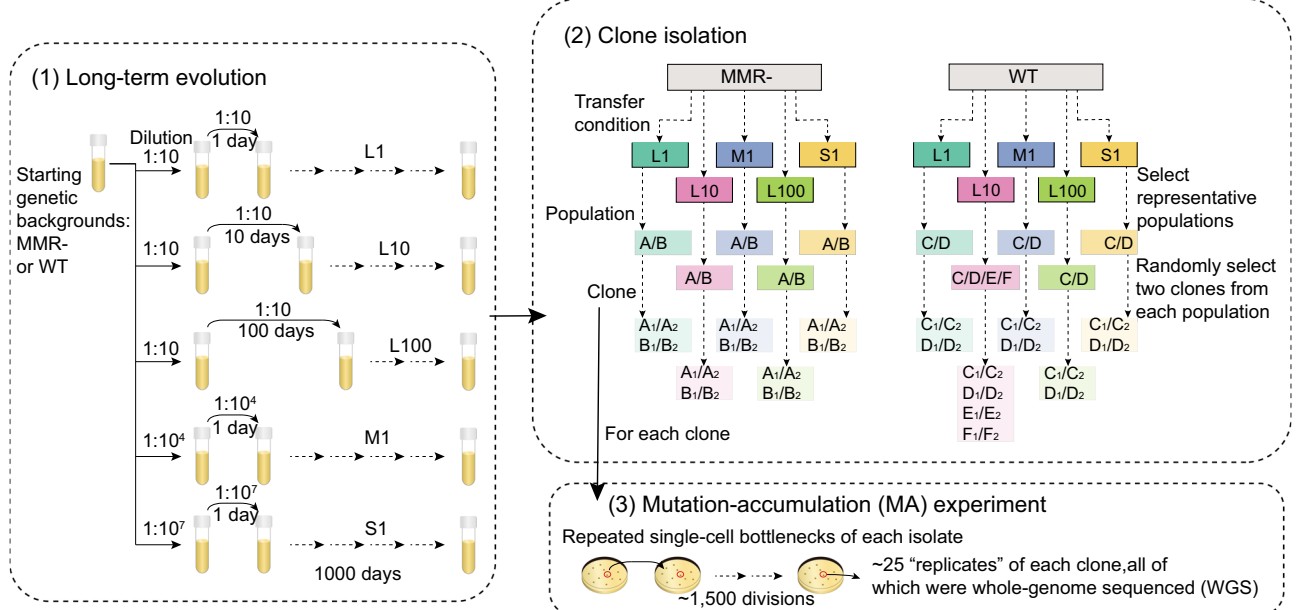

**Fig. 1 | Mutation-rate evolution experiment.** (1) Experimental evolution was run with two starting genetic backgrounds and five transfer schemes with various dilution factors and transfer periods (L1, L10, L100, M1, and S1) for 1000 days. (2) After experimental evolution, 44 clones were isolated from evolved populations. Specifically, we used two independent clones isolated from each of two

populations for each combination of genetic background and transfer scheme, except four independent populations for WT, L10 combination. A/B denote evolved populations with MMR- background; C, D, E, and F denote evolved populations with WT background. (3) Each isolated clone was subject to a MA/WGS experiment to estimate its rate and molecular spectrum of mutations.

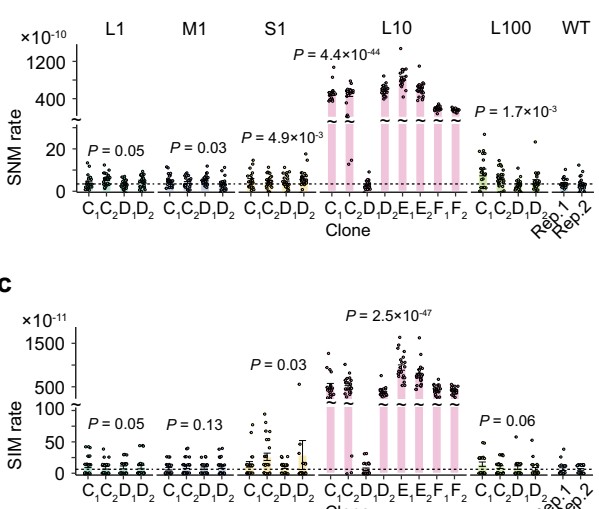

**Fig. 2 | Evolution of mutation rates.** Evolved single-nucleotide mutation (SNM) rates for (**a**) WT clones and (**b**) MMR- clones. $n = 25, 22, 25, 24, 24, 24, 23$, and 25 (L1-A1, A2, B1, B2, C1, C2, D1, and D2, respectively); $n = 23, 25, 22, 23, 25, 25, 24$, and 24 (M1-A1, A2, B1, B2, C1, C2, D1, and D2, respectively); $n = 21, 25, 22, 25, 22, 24, 25$, and 23 (S1-A1, A2, B1, B2, C1, C2, D1, and D2, respectively); $n = 18, 24, 24, 24, 19, 23, 24, 23, 23, 24, 22$, and 24 (L10-A1, A2, B1, B2, C1, C2, D1, D2, E1, E2, F1, and F2, respectively); $n = 24, 24, 24, 25, 22, 25, 23$, and 23 (L100-A1, A2, B1, B2, C1, C2, D1, and D2, respectively); $n = 21$ and 24 (WT); $n = 25$ and 24 (MMR-). Evolved small indel mutation (SIM) rates for (**c**) WT clones and (**d**) MMR- clones. $n = 25, 22, 25, 23, 24, 24, 24$, and 25 (L1-A1, A2, B1, B2, C1, C2, D1, and D2, respectively); $n = 23, 25, 22, 23, 24, 25, 24$, and 24 (M1-A1, A2, B1, B2, C1, C2, D1, and D2, respectively); $n = 20, 25, 23,$ 25, 22, 24, 25 and 24 (S1-A1, A2, B1, B2, C1, C2, D1, and D2, respectively); $n = 18, 24, 24, 24, 19, 24, 24, 23, 23, 24, 22,$ and 24 (L10-A1, A2, B1, B2, C1, C2, D1, D2, E1, E2, F1, and F2, respectively); $n = 24, 23, 26, 25, 22, 25, 23,$ and 23 (L100-A1, A2, B1, B2, C1, C2, D1, and D2, respectively); $n = 21$ and 23 (WT); $n = 24$ and 24 (MMR-). Data are presented as mean values +/– the standard errors of the means (S.E.M.); $P$-values were acquired by two-tailed unpaired $t$-tests contrasting MA lines from evolved populations and MA lines from the ancestor. Dashed lines represent absence of significant difference of mutation rates compared to the ancestral mutation rates. Rep.1 and Rep.2 denote two independent replicates from the same ancestral WT or MMR- progenitor. Source data are provided as a Source Data file (Data 1).

The aforementioned effect on the mutation-rate evolution in L10 populations, however, did not extend to the shortest or longest resource replenishment cycles. WT L100 clones exhibited only modest or insignificant increases in mutation rates, and their MMR- counterparts evolved significant reductions in SNM and SIM rates (Fig. 2). WT L1 clones also exhibited only modest and insignificant increases in mutation rates, and their MMR- counterparts evolved significant reductions in the SIM rate (Fig. 2). All of these data support theoretical expectations that hypermutators are most favoured in intermediately fluctuating environments[20].

**Mutation-rate evolution in response to population bottlenecks**
Contrary to the situation with L10 clones, clones from the two daily-transfer schemes with stronger population bottlenecks (M1 and S1) did not show highly increased mutation rates, and in the case of an MMR- background, they generally experienced reductions in their mutation rates after 1000 days of evolution. For example, the clones isolated from S1 populations show a significant reduction in the SNM rate by 41.6% ($P = 1.8 \times 10^{-8}$; Fig. 2b) and the SIM rate by 48.2% ($P = 4.2 \times 10^{-16}$; Fig. 2d). We also found that the SNM rate of the clone S1-A2 is much lower than the other clone from the same population, S1-A1, highlighting the intra-populational variation of mutation rates during the process of clonal evolution. For M1 populations with the MMR- background, the overall reduction is significant in the SNM rate (by 17.4%, $P = 0.01$; Fig. 2b) and the SIM rate (by 23.4%, $P = 6.1 \times 10^{-6}$; Fig. 2d), although the significance probably mostly comes from the clones of population M1-A. Therefore, the high mutation rate associated with the dysfunctional MMR is generally reduced after a long-term experimental evolution, supporting the idea that the overly high mutation rates can be deleterious to organisms.

In contrast, the evolved M1 and S1 clones isolated from populations with the WT background generally evolved increases in their mutation rates. Specifically, such clones evolved significant increases in SNM rates (M1: 1.3-fold, $P = 0.03$; S1: 1.5-fold, $P = 4.9 \times 10^{-3}$; Fig. 2a). In the WT background, there are increases of SIM rates in both M1 and S1 clones, and the increase in S1 clones is significant (3.1-fold, $P = 0.03$; Fig. 2c); the SVM rate in S1 clones is also significantly increased (3.9-fold, $P = 1.1 \times 10^{-3}$; Supplementary Fig. 2).

These observations indicate that the initial genetic background has a strong influence on the direction of mutation-rate evolution across population-size treatments. In WT populations, mutation-rate evolution is most likely to be driven by a reduction in the efficiency of selection on the mutation rate resulting from reduced $N_e$ or strong hitchhiking effects for hypermutators, which is not the case in MMR- populations. In fact, both MMR- and WT populations evolved to be fitter during the experimental evolution[25], but only WT populations evolved to have higher mutation rates. Therefore, the effect of increasing mutation rates was likely overwhelmed by much stronger selection against high mutation loads in MMR- populations.

**Genetic basis of emergent hypermutators in intermediate resource-replenishment cycles**
To gain insight into the potential molecular underpinnings of mutation-rate evolution, we tracked the genomic changes of the experimental populations through time. Some L10 WT populations may have fixed hypermutator alleles very early during experimental evolution, as the number of mutations in three of the eight assessed WT L10 evolved clones approached or even exceeded the number of mutations observed in MMR- L10 evolved clones (Supplementary Fig. 3). Using longitudinally-collected whole-population sequencing data[22,26], we identified four MMR-related mutations in seven L10

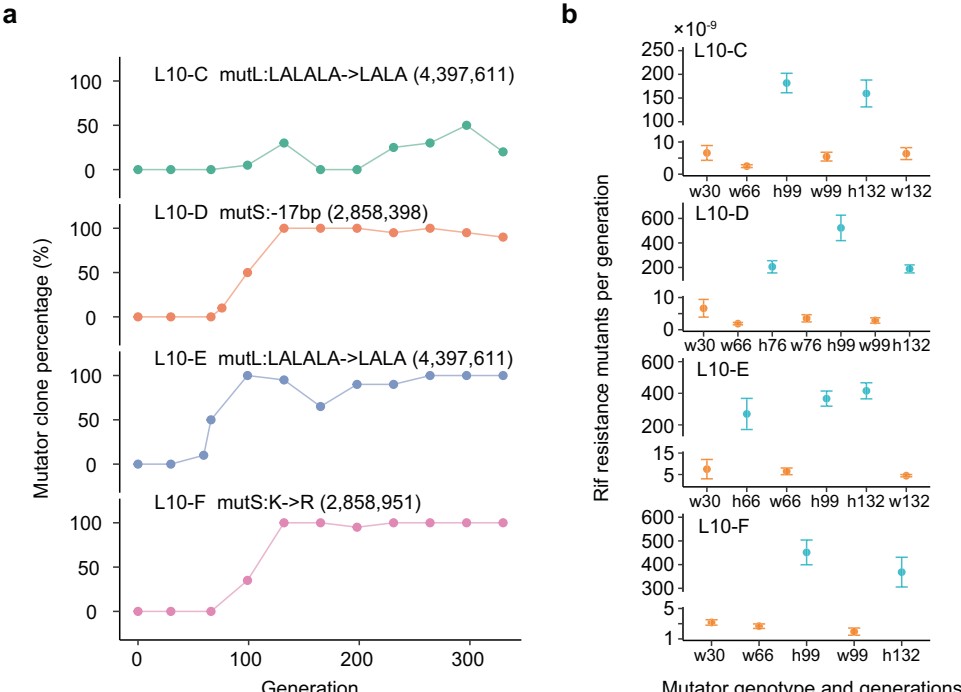

**Fig. 3 | Temporal evolution of hypermutation in the L10 transfer scheme.**
**a** Frequency trajectories of candidate mutations in each population. **b** Mutation rates of clones with different mutator genotypes isolated at different evolutionary times were estimated by fluctuation tests conferring rifampicin (Rif) resistance. $n = 3$ independent colonies per genotype and generations. Data are presented as mean values +/− S.E.M.. $X$-axis denotes the mutator genotype (w: wild-type, orange; h: hypermutator, blue) of the clone and the number of generations that the clone experienced in experimental evolution (e.g., h132 is a hypermutator-containing clone isolated from the sample that has experienced 132 generations of experimental evolution). Source data are provided as a Source Data file (Data 2).

evolved clones (L10-C1, C2, D2, E1, E2, F1, and F2) as candidate hypermutator alleles, including two independent cases of a 6-bp deletion that converted a triplet LALALA motif in MutL to a duplet LALA (populations L10-C and L10-E), a 17-bp deletion in the coding region of MutS (population L10-D), and a nonsynonymous substitution in MutS (population L10-F). Their earliest detected times are particulary short: only 99, 76, 59, and 99 generations for populations L10-C to F, respectively (Fig. 3a; Supplementary Fig. 4).

To infer the effects of these putative mutations on mutation rates, we performed fluctuation tests for rifampicin resistance on clones isolated from samples that had been frozen between 30 and 132 generations of experimental evolution, revealing that mutation rates had been elevated an average of 33-, 82-, 57-, and 158-fold in populations L10-C to F, respectively (Fig. 3b). On average, these putative mutations increased the mutation rates to rifampicin resistance 83-fold, which contributes to 89% of the averaged SNM and SIM mutation-rate increases for the entire experimental evolution. Overall, these results motivate the conclusion that increased mutation rates rapidly evolved in these populations. Notably, a previous study found that addition or subtraction of the LA repeat in the LALALA motif of MutL impairs the ATP-binding function critical for DNA mismatch repair[27].

### Hypermutator/antimutator accumulation affecting mutation spectra

The rise or fall of hypermutator or antimutator alleles also influences the spectra of mutations (Supplementary Fig. 5). Here, we focused on the contextual SNM profiles, counting the 96 possible classes of trinucleotide changes defined by the six outcomes for the focal nucleotide (C > A, C > G, C > T, T > A, T > C, and T > G mutations) and the identity of the immediate flanking 5′ and 3′ bases (Supplementary Fig. 6). Interestingly, most of the evolved populations with the MMR-background and the L10 WT populations with identified MMR-related

hypermutator alleles show elevated proportions of C > T mutations as compared to the MMR- ancestor (Supplementary Figs. 5, 6). We further applied a dimensionality reduction framework to extract the mutation patterns (MPs) from all evolved populations. The feature of elevated proportions of C > T mutations is shown in the first MP (Fig. 4a). Three distinct mutation patterns (MP1-3) were found across the MMR- populations and the L10 WT populations with identified MMR-related hypermutator alleles (Fig. 4a, b). These patterns were not observed in the other WT populations (Supplementary Fig. 7), suggesting that the mutational patterns observed in the L10 WT populations emerged with the arrival of hypermutator MMR alleles. MP1 and MP2 are correlated with mutational patterns found in other MMR- bacteria (Fig. 4c; Supplementary Fig. 8), as well as in MMR-defective cancers[28] (Fig. 4d; Supplementary Fig. 9).

## Discussion

In summary, using a comprehensive MA/WGS approach, we directly observed rapid evolution of mutation rates in haploid asexual populations responding to changes in population-genetic and ecological environments. These observations highlight the extraordinary bidirectional accessibility of mutation-rate affecting mutations. Of particular interest is the rapid emergence of reduced mutation rates in populations from which MMR was deleted and therefore incapable of reversion to WT. The emergence of antimutators on such a background suggests substantial excess capacity for mutation-rate reduction via other DNA replication and repair pathways[29,30] or still other mechanisms[31–33] (Supplementary Data 5). For example, a Pol III antimutator encoded by a *dnaE* allele can be effective in suppressing the loss-of-function of *mutT* and *mutL*[34]. This kind of situation is consistent with the hypothesis for the coevolution of the components of a layered surveillance system, as natural selection operates on the total error rate, and with multiple degrees of freedom, the enhanced efficiency of one component can result in the relaxed selection on others[35].

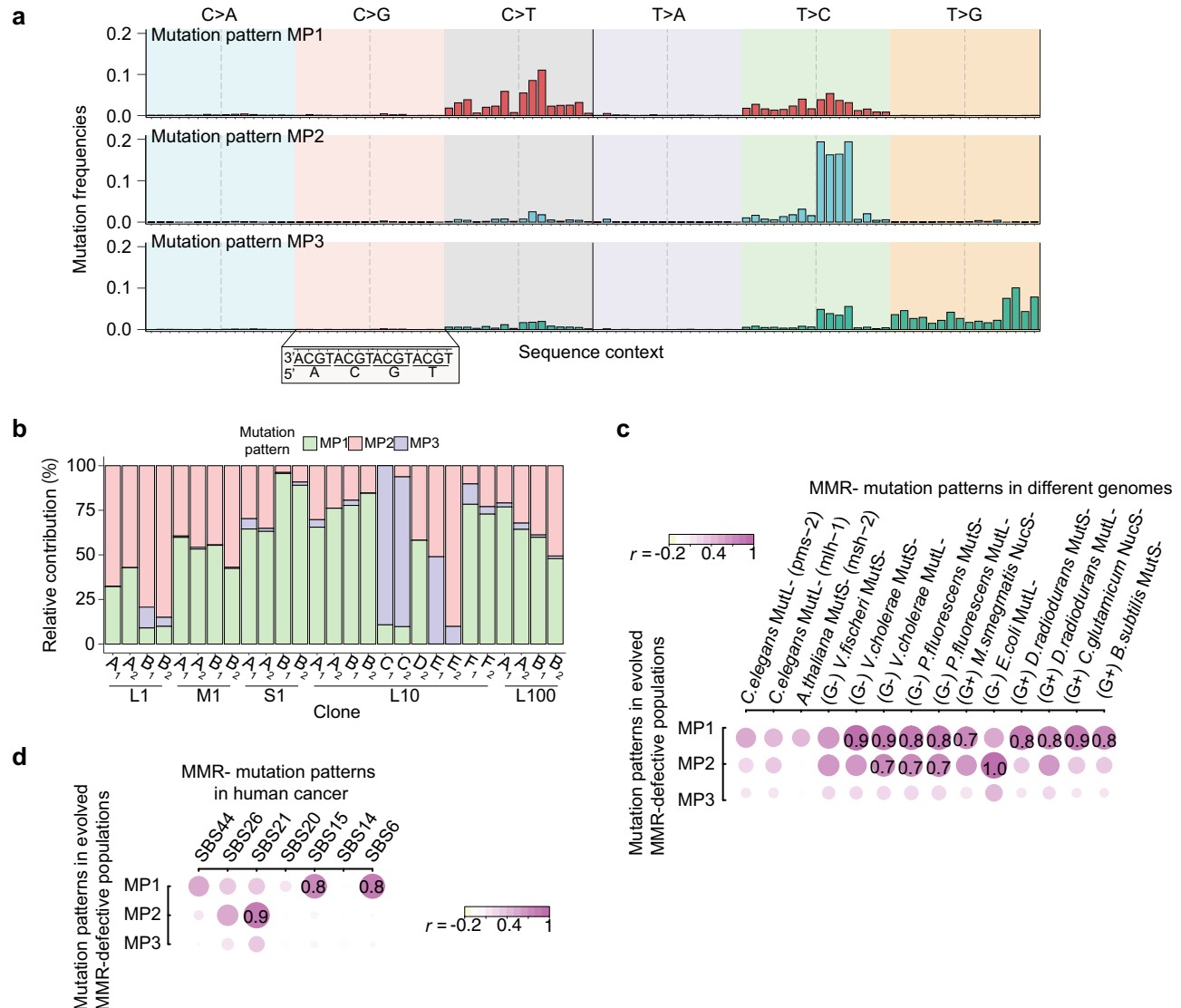

**Fig. 4 | Mutation patterns for MMR- populations. a** The 96-class contextual mutational spectra for the three mutation patterns (MPs) for the evolved populations with MMR- background or the L10 transfer scheme based on the dimensionality reduction analysis. Each bar represents a class; each class represents a single nucleotide change (one of six colored blocks; labels on the top) with the context of 3′ and 5′ flanking nucleotide (ticks at the bottom). **b** Three MPs largely contribute to the mutation profiles of clones isolated from the evolved populations with MMR- background or the L10 transfer scheme. **c** Correlation coefficients between each MP and each of 96-class mutational spectra from various MMR-deficient organisms (G+: Gram-positive bacteria; G-: Gram-negative bacteria). **d** Correlation coefficients between each MP and each of 96-class spectra for human cancer. SBS6, 14, 15, 20, 21, 26, and 44 are seven MMR-deficient spectra (single base substitutions; SBS) annotated in the human cancer database, COSMIC. For (**c**, **d**), the size and colour of circles represent the values of correlation coefficients (*r*; shown if ≥0.7). Source data are provided as a Source Data file (Data 3).

Across the Tree of Life, mutation rates vary by three orders of magnitude and inversely correlate with the effective population sizes of species, and this observation has been best explained by the drift-barrier hypothesis thus far[36–39]. Although other explanatory factors, such as body size, metabolic rate, and generation time[40,41] have been proposed, the focus in these studies has been primarily on animals, and all correlations are based on rates of molecular evolution rather than actual rates of mutation per cell division. Thus, even if these other factors show strong correlations with evolutionary rates, whether these traits can directly affect mutation-rate evolution is still questionable. For example, because organisms with larger effective population sizes tend to have small body sizes[42], an inter-specific correlation between body size and mutation rate is also consistent with the drift-barrier hypothesis.

Why is there a particularly high mutation rate in L10 populations but not in L100 populations? For the wild-type background, this non-monotonic relationship between mutation-rate evolution and the length of replenishing cycles cannot be fully explained by the drift-barrier hypothesis, as the $N_e$ reduction in L100 populations is presumably more severe than in L10 populations, evident by the slightly reduced population densities in L100 populations than in L10 populations[22]. Instead, we propose an explanatory model combining forces that lead to the evolution of reduced and elevated mutation rates. The forces causing the evolution of reduced mutation rates involve selection against high mutational loads. On the other hand, the forces causing elevated mutation rates involve hitchhiking effects with beneficial mutations, which are the strongest in fluctuating environments with intermediate cycle lengths[20,21]. As the overall elevating force is non-monotonic, and the overall reducing force does not prevail against the non-monotonicity of the elevating force, the combined outcome remains non-monotonic.

Except for the mutation-rate evolution in the intermediately fluctuating environment, many of our results are consistent with expectations of the drift-barrier hypothesis. First, mutation-rate evolved to be higher in smaller WT populations (S1 and M1) and to be lower in MMR- populations, consistent with the prediction that the drift-barrier of mutation rates is prominently influenced by demography. Second, as the selection against hypermutator alleles is not immediate, but only takes place after sufficient time has elapsed for the development of a linked load of deleterious mutations, a substantial wandering (at least several fold) of mutation rate over evolutionary time is predicted, with variance maintained within populations as well as over generations. As expected, the evolved clones show relatively high levels of within-population variation (e.g., an increased mutation rate for L10-D2 but not for D1), and parallel populations can evolve to different paths but the same direction on mutation-rate evolution (e.g., *mutL-* or *mutS-* in L10-C, D, E, and F).

A recent study analysed the mutation rates in natural and mutation-accumulated yeast strains and claimed to have falsified the drift-barrier hypothesis[31]. However, that work was based on indirect inference, not on experimental evolutionary analysis, and moreover relied on erroneous interpretations of the drift-barrier hypothesis. For example, their mutation-accumulated strains started with an MMR-background and restored the MMR gene to estimate the generated mutation-rate effects on a natural background. As the mutation spectra in MMR- and natural strains are different, the ratio of SIM to SNM produced in the experiment is dramatically higher than those in naturally evolving MMR-proficient strains that more likely define the historical evolution of the mutation rate. Importantly, the drift-barrier hypothesis for mutation-rate evolution predicts variation in mutation rate over evolutionary time, owing to drift around the drift barrier. In this recent study, 10 of the 13 isolates exhibited higher mutation rates than the ancestor, with the upward SNM rate variants having a ~30% inflation and the downward variants causing an average 23% reduction, which is consistent with the prediction of the drift-barrier hypothesis. Again, we emphasize that, to test such a theory more accurately, one probably needs to maintain experimental evolution under a constant environment for a longer time.

Analysing evolution alterations over just a 1000-day timespan, we observed how environmental fluctuations impact the dynamics of mutation-rate evolution, particularly in persistently fluctuating environments[43]. Future experimental and theoretical work that jointly incorporates the stochastic aspects of both finite-population size and environmental variation are highly desirable.

## Methods

### Long-term evolution experiments

All populations originated from two ancestral strains with different initial mutation-rates. Ancestral wild-type (WT) are descendants of PFM2, a prototrophic derivative of *E. coli* K-12 str. MG1655, while ancestral mismatch repair knockouts (MMR-) are descendants of PFM5 (PFM2, Δ*mutL*)[24]. The long-term cultures consist of three transfer-size treatments, differing by 1 mL (large transfer-size treatment, L1), 1uL (medium transfer-size treatment, M1) and 1nL (small transfer-size treatment, S1) daily transfer and maintained in 10 mL LB broth shaking at 175 rpm at 37 °C. In addition, the large transfer-size treatment was run with 10- (L10) and 100-day (L100) replenishment cycles to yield different starvation regimes. At each transfer condition, WT and MMR- were both applied with four parallel evolution experiments, generating $5 \times 2 \times 4 = 40$ populations. To maintain and trace the historical record of populations, 1 mL of copies of cultures were collected at days 0, 90 (or 100 for 100-day populations), 200, and then every 100 days, frozen with liquid nitrogen, and stored at −80 °C for sequencing. In addition, we froze samples every 20, 30, or 40 days for 1-day and 10-day populations until day 400 to revive populations in the case of contamination or extinction. Specifically, to avoid a case of a whole

population extinction in 100-day populations, we maintained 100-day populations in triplicate (1 main culture and 2 backup cultures) with only the main culture being used to seed new triplicate cultures every 100 days. One backup culture would replace the main culture if it extincted and the experiment would continue. In this study, whole-population extinction events only occurred during the first 200-days. During the experimental evolution, L10 and L100 populations did show smaller population sizes before the transfer than L1 populations, but the difference is small[22].

### Effective population sizes for L1, M1, and S1 populations

During the experimental evolution, we observed that the carrying capacities of the populations are always ~$10^{10}$ per tube. Theoretically, the effective population size is the harmonic mean of fluctuated census population sizes[44,45]. Therefore, for L1 populations, the dilution factor = $10^{-1}$, $N_e$ is about

$$\frac{5}{\frac{1}{10^9} + \frac{1}{2 \times 10^9} + \frac{1}{4 \times 10^9} + \frac{1}{8 \times 10^9} + \frac{1}{10^{10}}} \approx 2.5 \times 10^9$$

Similarly, for M1 populations, the dilution factor = $10^{-4}$, and $N_e$ is about $7.5 \times 10^6$. For S1 populations, the dilution factor = $10^{-7}$, and $N_e$ is ~$1.3 \times 10^3$.

### Rates of genomic evolution

We extracted DNA of long-term evolved populations from cultures every 100 days until day 900[22], and sequenced as paired-end reads to a target depth of 100× on an Illumina HiSeq 2500 or an Illumina NextSeq 500. Available sequencing profiles were identified using Breseq[46] from trimmomatic[47] trimmed reads. For each population, at each sample point, we quantified the genomic divergence by summation of derived allele frequencies of all detected SNM rates. Then, we performed linear regression by the function "lm" in R with formula "genomic divergence ~ guaranteed generations + 0" for estimating the rate of genomic evolution. The estimated generations per day are 3.3, 13.3, 23.3, 0.33, and 0.25 for L1, M1, S1, L10, and L100 populations, respectively[22,25].

### Mutation accumulation (MA) procedure

We labelled MMR- parallel populations as A or B, and WT populations as C, D, E or F. After 1000 days of evolution, each combination of transfer conditions and initial mutation rate was assessed in quadruplicate with two clones assessed from each of two representative populations, except for WT L10, where two additional populations were also investigated because they exhibited extremely high evolutionary rates and fluctuation tests suggested these populations may have evolved uniquely high mutation rates (Supplementary Fig. 1). In total, we focused on 44 experimental clones (2 long term evolution initial mutation rates × 5 transfer conditions × 2 populations × 2 clones + 4 extra WT L10 clones = 44) and 4 ancestral clones (2 WT and 2 MMR-). Each clone was cultured on LB (Miller) agar plates, and 50 distinct colonies were obtained to form parallel MA lines. These MA lines were streaked onto new plates from random single colonies daily for up to 60 days; this transfer method strongly bottlenecks the MA lines so that nearly all mutations are fixed though genetic drift, regardless of their fitness effects[24]. The offspring of three clones (L10-A1, L10-A2 and L100-C1) grew so slowly that transfers were done every other day and had to be terminated early (30, 31, and 38 days).

The number of generations ($G$) per passage were estimated by CFU counts of dissolved and diluted colonies of focal plates as follow:

$$G = \frac{(G_0 + G_{30}) \times a + (G_{30} + G_N) \times b}{N}, (a = 15; b = N/2 - 15), \quad (1)$$

where $G_0$, $G_{30}$ and $G_N$ are number of generations respectively estimated at days 0, 30, and the last day of the entire MA experiment.

Typically $G$ ranges from 18 to 24. $N$ is number of days of the entire MA experiment.

At the end of MA procedure, WGS was applied to these 48 evolved and ancestral MA progenitor clones and ~1200 of their derived MA lines (~25 MAs per clone) after propagation for an average of ~1500 generations.

## Genomic DNA isolation and high-throughput sequencing

Genomic DNA samples of MA ancestral clones and MA lines were extracted using the Wizard DNeasy UltraClean Microbial Kit (Qiagen), Wizard Genomic DNA Purification Kit (Promega), or the MagMax DNA Multisample Ultra 2.0 Kit (Applied Biosystems). Fresh DNA from MA lines or progenitor clones was submitted to the Beijing Genomics Institute (BGI-Hongkong) and sequenced on a DNBseq-G400 platform.

## Variants calling

Using trimmomatic[47], we removed residual adapters and trimmed low quality sequences. Cleaned reads were then mapped to the reference genome of *E. coli* K-12 str. MG1655 (NC_000913.3) with BWA[48]. Only paired reads were allowed to map. After the mapping, 15 MA lines were discarded from the following analysis because of their low depths (<50×) or low coverages (<30%), which are considered as contaminated lines during the MA experiments. In the end, on average, 99.6% of genomic positions were covered by the high quality reads (Q30), displaying an average depth of 149×.

We then removed duplicated reads from filtered alignments, and called candidate SNM and SIM using GATK4[49]. Base quality scores were corrected by corresponding MA progenitor. The candidate variants with read depths <4 or phred scores <10 were discarded. Ancestral variants were also discarded; the ancestral variants were identified from the differences between the reference genome and the MA progenitors, which are expected to be supported by at least four reads with phred scores higher than ten. In addition, only the indels <= 4 bp (SIM) were retained. The variants that were consensus among MA lines were also discarded; consensus SNM calls (or SIM) were identified if present in more than two (or four) MA lines with the same MA ancestor. To detect structural variations (SVs), including deletions, duplications, inversions, and insertions, we used CNVnator[50], Lumpy[51] and BreakDancer[52]. The redundant predictions were discarded. Among the maximum 500 bp upstream and downstream extended regions of the predicted SVs, we first removed the redundant predictions of CNVnator from the Lumpy predictions. We then removed the redundant predictions of Breakdancer if they are present in the Lumpy/CNVnator predictions. For each MA experiment, we removed SVs that were either called in the MA ancestral clone or found in at least three parallel MA lines, allowing a maximum 500 bp extension or substraction at both ends.

## Estimating mutation rates

SNM and SIM rates (per genomic site per generation) were calculated for each MA line by the equation:

$$u = \frac{m}{G \times n}, \tag{2}$$

where $u$ denotes the mutation rate, $m$ represents the number of SNM or SIM observed, $G$ is the total number of generations elapsed per lineage, and $n$ is the number of ancestral sites covered by sequencing reads with high quality.

SVM rates (per genome per generation) were calculated for each MA line by the equation:

$$u = \frac{m}{G}, \tag{3}$$

where $m$ represents the observed SV numbers.

## Excluding outliers with odd mutation rates

Before analyzing SNM rates, we removed several MA lines with odd rates using a cutoff of absolute $Z$-score > 2.5:

$$Z = \frac{u - U}{S}, \tag{4}$$

where $u$ is the mutation rate of a MA line, $U$ and $S$ are the mean and standard deviation of mutation rates for all lines related to the same MA ancestor. We also adopted this cutoff for filtering odd SIM and SVM rates in analysis. All filtered mutations in MA experiment are listed in Supplementary Data 2–4.

## Mutation trajectories

To find candidate mutations that associate with mutation-rate evolution, we investigated nonsynonymous substitutions and indels in MA ancestors, including variants called from ancestral clones and consensus calls among MA lines with the same MA ancestor (see above mentioned). Here, we focused on genes involving DNA replication and repair processes, including GO-terms of "0006298," "0006281," "0006260," "0006285," "0006284," "0006307," "0000731," "0007059," "0051103," "0006310," "0009432," "0045004," "0006261," and "0006271" (http://geneontology.org/). The temporal trajectories of allele frequencies for these putative mutations were directly from the detected mutations of previous longitudinal sequencing data following previous methodology (Supplementary Fig. 4)[22].

## Polymerase chain reactions (PCR)

To validate mutational trajectories of candidate hypermutators in *mutL* or *mutS*, we designed primers targeting the mutated regions using Primer-BLAST (https://www.ncbi.nlm.nih.gov/tools/primer-blast/). The targeted PCR products are 900 and 700 bp for *mutS* and *mutL*, respectively. Below are the primer sequences used.

*mutS*-F: 5′-TCTGGCACGTCTGGCTTTAC-3′;
*mutS*-R: 5′-AGATGCGATCGATAGGTCCAA-3′;
*mutL*-F: 5′-TCAGGTCTTACCGCCACAAC-3′;
*mutL*-R: 5′-GCCAGCGCTTGTTCAAGAAA-3′.

To estimate the allele frequency of candidate hypermutators during the experimental evolution, we streaked frozen stocks for WT-L10-C, D, E, and F onto LB plates and cultured overnight at 37 °C. We used frozen samples collected at days 0, 90, 180, 200, 230, 300 and then every 100 days. To prepare samples for PCR, we picked 20 single colonies from each stock randomly and then transferred each of them to a 1.5 mL tube containing 0.6 mL of LB broth for overnight culture. Following overnight incubation, 1 uL of each revived colony was taken to perform PCR using Q5 High-Fidelity 2X Master Mix (NEB). The PCR conditions were starting with 1 min at 98 °C for denaturation, then subjected to 30 cycles of amplification (10 s at 98 °C for denaturation, 30 s for annealing, and 1 min at 72 °C for extension). The annealing temperatures are 66 °C for the *mutS* primers and 68 °C for the *mutL* primers. At the end, a final extension was performed 2 min at 72 °C. PCR products were then purified and collected by GeneJET PCR Purification Kit (Thermo Fisher Scientific). Purified DNAs were submitted to the KED Genomics Core at Arizona State University for Sanger sequencing.

By testing PCR products of 20 single colonies from each frozen stock, we found that several colonies from as early as day 300 of WT-L10-C and WT-L10-F, day 230 of WT-L10-D, and day 180 of WT-L10-E contain hypermutators. Thus, the occurrence times of hypermutator candidates were estimated to be between days 230–300, 200–230, 90–180, and 230–300 for WT-L10-C, D, E, and F, respectively.

## Fluctuation test

Luria-Delbrück fluctuation tests can be used for the determination of the spontaneous mutation rate using a known molecular marker, such

as antibiotic resistance[53]. To study whether the candidate hypermutators affect mutation rates, we used the fluctuation test with rifampicin resistance for clones isolated from L10-C, D, E and F at days 90, 200, 300, and 400, as well as the MA ancestor. Whether the clones with or without the candidate was examined by PCR as mentioned above. Three independent clones with the hypermutator and/or three independent clones without the hypermutator were used for each L10 population at each time-point.

During the fluctuation test, each targeted clone was first overnight cultured, and 44 replicates of the targeted clones were then transferred (1: $10^7$ dilution) and grown for 24–48 h in a 96-well culture plate containing 1 mL of LB broth per well. Four were diluted and spread on LB agar for total cell counts; 40 were plated on LB + Rifampin agar (100 mg/L) for resistance selection. The number of resistant mutants in CFU/mL were converted to an estimated mutation rate by the "newton.LD" function in the R package "rSalvador"[54].

### Mutation patterns in populations

In addition to the six-class SNM spectra (C > A, C > G, C > T, T > A, T > C, and T > G), we also reported the mutation contexts with 96 classes constituted by six SNM classes, plus the flanking 5' and 3' A, G, C, and T bases. To further extract orthogonal mutation patterns from the evolved populations, we constructed mutation tables with 96-class mutation contexts and identified the "mutation patterns (MPs)" using the non-negative matrix factorization (NMF) function in the R package "MutationalPatterns"[55]. Following the recommendation in the manual, to determine the optimal number of MPs, we estimated cophenetic correlation coefficients with different MP numbers. We choose the smallest values for which cophenetic correlation coefficient starts decreasing as the best number of MPs. We finally obtained three MPs related to MMR-defect populations and one MP for WT populations.

### MMR-defect mutation patterns in other species

We obtained mutations from several MMR deficiency related MA studies, including *Bacillus subtilis* NCIB 3610[56] (NZ_CM000488.1), *Corynebacterium glutamicum* ATCC 13032[57] (NC_003450.3), *Deinococcus radiodurans* R1[58,59] (NC_001263.1, NC_001264.1), *Mycobacterium smegmatis* MC2 155[60] (NC_018289.1), *Pseudomonas fluorescens* SBW25[59] (NC_012660.1), *Vibrio Cholerae* 2740-80[61] (NZ_CP016324.1, NZ_CP016325.1) and N16961[62] (NC_002505.1, NC_002506.1), and *Vibrio Fischeri* ES114[61] (NC_006840.2, NC_006841.2). All bacterial genomes were downloaded from NCBI and used to estimate 96-class spectra. The MMR-defect mutation spectrum of *Caenorhabditis elegans* was obtained from ref .[63]. The mutation data for *Arabidopsis thaliana* MMR- lines were obtained from ref. [64], and we downloaded the reference genome from The Arabidopsis Information Resource (https://www.arabidopsis.org/) to estimate 96-class spectra.

### Cancer mutation patterns

A number of Single Base Substitution (SBS) patterns extracted from human cancer samples were obtained from COSMIC (https://cancer.sanger.ac.uk/cosmic/signatures), seven of which are associated with defective DNA mismatch repair: SBS6, SBS14, SBS15, SBS20, SBS21, SBS26, and SBS44[28]. These mutation patterns are involved in 67, 33, 220, 83, 13, 37, and 246 cancer genomes across 13, 4, 12, 7, 6, 11, and 11 cancer types, respectively.

### Reporting summary

Further information on research design is available in the Nature Research Reporting Summary linked to this article.

## Data availability

The MA/WGS data generated in this study have been deposited in the NCBI's short read archive database under PRJNA688002. The *Arabidopsis thaliana* genome used in this study is available in the The Arabidopsis Information Resource database [https://www.arabidopsis.org/]. Other genomes are available in the NCBI Genbank under NC_000913.3, NZ_CM000488.1, NC_003450.3, NC_001263.1, NC_001264.1, NC_018289.1, NC_012660.1, NZ_CP016324.1, NZ_CP016325.1, NC_002505.1, NC_002506.1, NC_006840.2, and NC_006841.2 [https://www.ncbi.nlm.nih.gov/genbank/]. GO-terms are available at gene ontology database [http://geneontology.org/]. SBS patterns of human cancer samples are available in COSMIC database [https://cancer.sanger.ac.uk/cosmic/signatures]. Source data are provided with this paper.

## Code availability

The code is available upon request.

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

## Acknowledgements
We thank Ryan Stikeleather, Stephan Baehr, and Jakob Carlson for their assistance and helpful comments. This work was supported by the US Army Research Office Multidisciplinary University Research Initiative (MURI) awards W911NF-09-1-0444 and W911NF-14-1-0411, the National Institutes of Health Maximizing Investigators' Research Award (MIRA) R35-GM122566-01, and Natural Science Foundation of Chongqing cstc2019jcyj-msxmX0099.

## Author contributions
W.W., W-C.H., M.G.B., S.F.M., and M.L. designed research; W.W., S.F.M., and G.B. performed experiments; W.W. and W-C.H. conducted bioinformatic analyses; W.W. and W-C.H. analyzed data; and W.W., W-C.H, M.G.B., and M.L. wrote the paper.

## Competing interests
The authors declare no competing interests.
