## [Peer Review File · Nature Communications]

Rapid evolution of mutation rate and spectrum in response to environmental and population-genetic challengesReviewers' Comments:

Reviewer #1:

Remarks to the Author:

This is an exceptionally interesting and original work on an important subject. To my knowledge, this is the first attempt to directly test the drift-barrier hypothesis (DBH) on mutation rate using state-of-the-art toolkits of experimental evolution and mutation-accumulation (MA) assays. My main concerns are about the interpretation of the results.

As described by the authors, DFB postulates that "lower mutation rates are favored by natural selection due to the prevalence of deleterious mutations, with the power of natural selection to reduce error rates being constrained by the magnitude of genetic drift defined by the prevailing N_e ". Accordingly, in its simplest form, the hypothesis assumes that elevated mutation rate has no direct selective advantage (or this advantage is only temporary). However, as also very clear to the authors, several prior works showed that elevated mutation rate readily evolves in the laboratory in response to nutrient limitation or other forms of external stress. As mutator alleles speed up adaptation, they hitch-hike with the adaptive mutation they generated in asexual populations.

In my eyes, rigorous testing of DFB would require experimental conditions where positive selection on mutator alleles is minimized, i.e. when stabilizing selection dominates on most alleles in the genome. This would require control experiments that demonstrate that the starting strain is well-adapted to the nutrient-rich conditions (or long-term starvation) applied during the course of laboratory evolution. If it were not so, adaptation to the external conditions could drive the fixation of mutator alleles in the population.

Therefore, it is not surprising that long-term starvation (as a major stress factor) selects for an increased mutation rate, while changes in mutation rate are relatively minor under nutrient-rich conditions. In my eyes, the long-term starvation experiments do not directly test the DFB.

Probably the most exciting aspect of the work is the population genetic conditions under which the mutation rate is reduced. This could be directly relevant for DFB. However, it would be important to demonstrate directly that antimutators have been positively selected and they indeed improve fitness.

In a similar vein, it is unclear to me how N_e should affect the evolution of mutator alleles when the population is under strong stress (e.g. long term starvation). Under low N_e , the mutational supply rate is low, and therefore selection for mutators should be especially beneficial. However, the converse might also be true, as high N_e increases the efficiency of selection. I'm sure prior population genetic models have investigated this issue, but I doubt that such considerations would be directly relevant to DFB.

Reviewer #2:

Remarks to the Author:

Rapid evolution of mutation rate and spectrum in response to environmental and population-genetic challenges, by W. Wei et al., submitted to Nature Communications.

This group reports on a very large amount of work, and they are to be commended for taking this on. Their interest is whether and how mutation rates in unicellular organisms like bacteria can evolve over time. As part of this, they first set out to test the Drift-Barrier hypothesis (DBH) for mutation rates. In this hypothesis, mutation rates are thought to generally trend lower, as lower rates will reduce the incidence of deleterious mutations, but this downward trend is counteracted by the phenomenon of

genetic drift, which presents a barrier to going down further. Importantly, the impact of genetic drift is thought to be a function of the population size: in large populations genetic drift is a relative minor force, leading to lower overall mutation rates, while in smaller populations drift becomes more dominant, potentially producing higher mutation rates. The authors test this by growing *E. coli* cultures for 1,000 consecutive days with daily transfers to fresh growth medium, with the transferred volume varied from large to small, thus creating a set of different experimental population sizes. In addition, they also include conditions in which they subject the larger populations to less frequent refreshments (every 10 days, or every 100 days) to see if prolonged conditions of nutrient starvation also may affect the mutation rate. At the end of the long-term transfers they measure the evolved rate via MA (Mutation Accumulation) using two individual colonies from the final populations, and compare their rate to that of the ancestral line. For all these experiments, both wild-type and mutator variant of *E. coli* defective in mismatch repair are run in parallel, and this allows the measurements to be performed, at least in principal, at low and high starting mutation rates.

This work yields some interesting observations. For the wild-type strain, small (1- to 2-fold) increases in mutation rate are seen after 1,000 days for all three population sizes, in a manner that the authors consider consistent with the Drift Barrier hypothesis (largest effect for the smallest population size). However, for the MMR- mutator strain the opposite is seen, a generally downward trend for the rate with decreasing population size. More dramatic results are seen with the nutrient-starved cultures. For cultures starved for 10-day periods, several of them show strongly increased mutation rates, as if resembling the emergence of new or additional mutator phenotypes. Some evidence is presented is that, at least in some cases, a MMR-defective mechanism is responsible. However, unexpectedly, cultures starved for 100-day periods do not show these effects, and in many cases they reflect a decrease in the mutation rate.

Overall, this work is valuable and raises intriguing questions. However, it does not feel like a sufficiently strong new picture emerges from this work that would provide for a more detailed or comprehensive understanding of the forces that drive the mutation rate. At best, the current picture remains fragmentary. While the authors state that the wild-type data are consistent with the DBH, these effects are mostly very small. It may also be that the comparison to the ancestral rate is limited by the fact that there are only two ancestral values to compare to (one would have liked to see several more to allow a fair comparison between evolved and non-evolved rates). In contrast to the wt data, the MMR- data do not conform to the DBH hypothesis and, in fact, move in the opposite direction. So, a consistent pattern on the DBH has not emerged. Other observations that limit the reaching of overarching conclusions from this work is that the results for the 10-day starvation regime differ dramatically from those of the 100-day starvation. Ad hoc explanations for all of these diverse findings can be advanced, as the authors do, and they do not sound unreasonable, but all this does not help with formulating consistent models that might be helpful to readership. Finally, even for the two parallel cultures that make up a given treatment group, the results are often quite dissimilar. For example, clones A and B for the MMR- populations (Fig. 1B) are very dissimilar, the same for clones C and D (Fig. 1D) for the wt populations. Even within one clone, several disparate results are observed. For example, see A1 and A2 for MMR- strain (Fig. 1B) or D1 and D2 for the wt strain (Fig. 1D). All of this can best be summarized by saying that evolution in individual cultures can follow different paths and that similar outcomes are often not to be expected. This is, in principle, very interesting and intriguing, and constitutes impetus for all kinds of further explorations. However, in my view, the current picture that emerges here is too fragmentary and does not provide a sufficiently coherent message that would be of interest to a wider readership.

One other important point:

For the L10 and L100 experiments, no information is given about the health or sustainability of the cultures during and after the 10- or 100-day starvation periods. It is my personal experience that *E. coli* cultures after they reach stationary phase start to decline in viable titer if maintained at 37C within days. There is also a considerable amount of literature on this subject. To be honest, I would be

surprised if the cultures as used here would not have suffered many logs of viability loss. This type of information should have been provided. Any killing would obviously have consequences for the experimental outcomes as they reduce the effective population size. This may well provide an explanation for the intriguing differential L10 and L100 results, as the effects on population size may be dramatically different for the two conditions. This size difference may allow expression of mutator alleles in one case but not the other.

Reviewer #3:

Remarks to the Author:

Review of "Rapid evolution of mutation rate and spectrum in response to environmental and population-genetic challenges" by Wei et al.

This interesting manuscript describes a simple and important experiment: testing the evolution of mutation rates in *E. coli* during 1000 days of evolution under different setups of selection, population size, and baseline mutation rate. The results are very interesting and open additional questions: mutation rates changed significantly during the experiment, either up (in most wildtype populations) or down (in most MisMatch Repair populations). Population size (the dilution factor) had a minor effect, but the effect of selection was dramatic and non-monotonous: L10 groups (transfer every 10 days) showed significantly elevated mutation rates in most groups, in comparison with both L1 and L100 groups (transfer every 1 and 100 generations, respectively).

I think the manuscript is important and interesting, but its presentation requires revision. A broader context of different theories for the evolution of mutation rates should be included, and the results need to be discussed in that context.

Specific comments:

Introduction

p. 1, Lines 4-6 "Across the Tree of Life, mutation rates vary by three orders of magnitude, and are inversely correlated with the effective population sizes (N_e) of species 5,6. Thus far, this observation has been best explained by the drift-barrier hypothesis (DBH)"

A broader context should be included. Multiple factors are commonly suggested (body size, generation time, metabolic rate, environmental changes), and some of them have complex interactions with population size (e.g. large body size is correlated with small effective population size).

Abstract and Discussion

Abstract : "These results ... are broadly consistent with the drift-barrier hypothesis for the evolution of mutation rates". This claim requires further justification in the body of the manuscript.

p. 8 ln 14 "our results are consistent with many of the expectations of the DBH"

Which expectations of the DBH are the results consistent with? Please elaborate. In fact, the naïve reader may consider the interesting results of the paper as examples for the role of selection and environmental changes in the evolution of mutation rates (compare L1, L10, L100), while the population size appears to play only a minor role here (comparison of L1, M1, S1). A discussion of what theories would best explain the results is missing, and I think it could contribute to the paper.

p. 4 lines 18-21 "These outcomes may be a consequence of a reduction in the efficiency of selection on the mutation rate resulting from increased genetic drift under N_e reduction in WT populations, but an offsetting of this effect by selection in MMR- backgrounds that generate extremely high mutation

loads”

This is one possible explanation. Can the authors discuss other explanations as well? could selection, for example, be responsible for the increased mutation rate in WT populations of all sizes in a rapid growth setup?

P.4 In 23 – p.5 lines 1-2 “Although a recent study claims to have falsified the DBH [23], that work was based on indirect inference, not on experimental evolutionary analysis, and moreover relies on erroneous interpretations of the predictions of DBH theory.”

If the authors want to discuss this disagreement, additional details of [23] and the relevant predictions of DBH are needed.

p. 5, lines 15-end of paragraph

“Why is there a particularly high mutation rate in L10 populations but not in L100 populations?...”

This is a very puzzling question, and a deeper investigation of it may contribute to the paper. The explanation suggested by the authors involves “fluctuating environment with intermediate cycle length”. However, several parameters are different between L10 and L100: the stress is more dramatic in L100 (mentioned by the authors – but by how much? Can the authors evaluate? and may the stress affect mutation rates?), the number of generations is longer in L10 (by how much?), and of course the period between changes is indeed longer in L100. How are these factors combined? A model could perhaps help here, or at least a deeper discussion of the different factors and their interaction.

Figure 2

The caption of figure 2 is very short and minimalistic, and the text does not refer to Fig. 2 in detail (only in p. 3 line 3 “(Figs. 1B-E, 2; Extended data Table S1; Extended data Fig. 4 S2)”). Which of the results detailed in the text are presented in Fig. 2? Please also compare Fig. 1B-E and Fig. 2.

RESPONSE TO REVIEWER COMMENTS

Reviewer #1 (Remarks to the Author):

This is an exceptionally interesting and original work on an important subject. To my knowledge, this is the first attempt to directly test the drift-barrier hypothesis (DBH) on mutation rate using state-of-the-art toolkits of experimental evolution and mutation-accumulation (MA) assays. My main concerns are about the interpretation of the results.

Response: We thank the reviewer for acknowledging the originality of our work.

As described by the authors, DFB postulates that “lower mutation rates are favored by natural selection due to the prevalence of deleterious mutations, with the power of natural selection to reduce error rates being constrained by the magnitude of genetic drift defined by the prevailing N_e ”. Accordingly, in its simplest form, the hypothesis assumes that elevated mutation rate has no direct selective advantage (or this advantage is only temporary).

Response: We agree with the definition of the drift barrier hypothesis here.

However, as also very clear to the authors, several prior works showed that elevated mutation rate readily evolves in the laboratory in response to nutrient limitation or other forms of external stress. As mutator alleles speed up adaptation, they hitch-hike with the adaptive mutation they generated in asexual populations.

Response: Correct, stressful environments have been shown to lead to elevated mutation rates, and in the revised manuscript, we emphasize this further in the introduction. However, prior experimental-evolution studies usually assume a novel and simple environment with a single adaptive optimum. Adaptation to environments with high heterogeneity remains less explored, and may result in distinct patterns of mutation-rate evolution, as reported in this manuscript. We have also modified the sentences in the Introduction to emphasize this point.

In my eyes, rigorous testing of DFB would require experimental conditions where positive selection on mutator alleles is minimized, i.e. when stabilizing selection dominates on most alleles in the genome. This would require control experiments that demonstrate that the starting strain is well-adapted to the nutrient-rich conditions (or long-term starvation) applied during the course of laboratory evolution. If it were not so, adaptation to the external conditions could drive the fixation of mutator alleles in the population.

Response: The reviewer states that the drift-barrier hypothesis would be the best validated under the situation where stabilizing selection is dominant, and positive selection is minimized. Although we agree with the reviewer’s statement, we think this situation is rarely met for the natural populations. Instead, in this work, we are more interested in testing how mutation-rate evolution can be influenced by the breadth of environmental and demographic changes that natural populations may frequently encounter, not simply on the drift barrier. For these more complex environmental settings, we are testing the applicability of the drift-barrier hypothesis and alternative hypotheses, such as the hitchhiking-during-adaptation hypothesis.

To make this more clear, we have substantially edited the Introduction to elaborate on the focal question in this manuscript.

Therefore, it is not surprising that long-term starvation (as a major stress factor) selects for an increased mutation rate, while changes in mutation rate are relatively minor under nutrient-rich conditions. In my eyes, the long-term starvation experiments do not directly test the DFB.

Response: Although it has been found that higher mutation rates can evolve in different stressful environments, the past literature and theory are more focused on the challenge of single environments. Instead, our manuscript addresses an environment with more temporal heterogeneity. More importantly, our observation that L10 instead of L100 evolved the highest mutation rates suggests that the fluctuation of feast and famine may be a more effective driving force than the near continuous stress of starvation, i.e., the L10 treatment represents the most variable environment. Thus, the drift-barrier hypothesis may be less relevant for evolution in fluctuating environments, as the reviewer notes. In the revised manuscript, we have elaborated more on the lack of empirical support for the evolution of higher mutation rates in fluctuating environments in the Introduction section and the Results section.

Probably the most exciting aspect of the work is the population genetic conditions under which the mutation rate is reduced. This could be directly relevant for DFB. However, it would be important to demonstrate directly that antimutators have been positively selected and they indeed improve fitness.

Response: We agree that the emergence of antimutators is an exciting observation and highly relevant to the drift-barrier hypothesis. However, it is extremely challenging to determine the specific molecular features of the emergent antimutators, as there are the number of potential antimutators that emerged early in the MMR- lines is very high. We now supplement the list of candidate antimutators at the end of manuscript and hope the progress of finding responsible antimutators can be accelerated (Supplementary Table 5). We plan to investigate the fitness effects of candidate antimutators in the future work, but this can only be done if the causal mutation(s) can be identified and isolated on a fixed genetic background, devoid of the remaining crop of mutations.

In a similar vein, it is unclear to me how N_e should affect the evolution of mutator alleles when the population is under strong stress (e.g. long term starvation). Under low N_e , the mutational supply rate is low, and therefore selection for mutators should be especially beneficial. However, the converse might also be true, as high N_e increases the efficiency of selection. I'm sure prior population genetic models have investigated this issue, but I doubt that such considerations would be directly relevant to DFN.

Response: As we observed elevated mutation rates in L10 populations, we propose that effective population size is likely less relevant to this result. In particular, as compared to L1, both L10 and L100 experienced more stress and reductions of population size (ref.22, Extended Data Figure 7), but the results are different for their mutation-rate evolution. The drift-barrier hypothesis alone cannot fully explain why L10 populations evolved higher mutation rates than L100 populations. We added more discussion on how N_e can impact mutation-rate evolution under environments with fluctuating resources.

Reviewer #2 (Remarks to the Author):

Rapid evolution of mutation rate and spectrum in response to environmental and population-genetic challenges, by W. Wei et al., submitted to Nature Communications.

This group reports on a very large amount of work, and they are to be commended for taking this on. Their interest is whether and how mutation rates in unicellular organisms like bacteria can evolve over time. As part of this, they first set out to test the Drift-Barrier hypothesis (DBH) for mutation rates. In this hypothesis, mutation rates are thought to generally trend lower, as lower rates will reduce the incidence of deleterious mutations, but this downward trend is counteracted by the phenomenon of genetic drift, which presents a barrier to going down further. Importantly, the impact of genetic drift is thought to be a function of the population size: in large populations genetic drift is a relative minor force, leading to lower overall mutation rates, while in smaller populations drift becomes more dominant, potentially producing higher mutation rates. The authors test this by growing *E. coli* cultures for 1,000 consecutive days with daily transfers to fresh growth medium, with the transferred volume varied from large to small, thus creating a set of different experimental population sizes. In addition, they also include conditions in which they subject the larger populations to less frequent refreshments (every 10 days, or every 100 days) to see if prolonged conditions of nutrient starvation also may affect the mutation rate. At the end of the long-term transfers they measure the evolved rate via MA (Mutation Accumulation) using two individual colonies from the final populations, and compare their rate to that of the ancestral line. For all these experiments, both wild-type and mutator variant of *E. coli* defective in mismatch repair are run in parallel, and this allows the measurements to be performed, at least in principal, at low and high starting mutation rates.

This work yields some interesting observations. For the wild-type strain, small (1- to 2-fold) increases in mutation rate are seen after 1,000 days for all three population sizes, in a manner that the authors consider consistent with the Drift Barrier hypothesis (largest effect for the smallest population size). However, for the MMR- mutator strain the opposite is seen, a generally downward trend for the rate with decreasing population size. More dramatic results are seen with the nutrient-starved cultures. For cultures starved for 10-day periods, several of them show strongly increased mutation rates, as if resembling the emergence of new or additional mutator phenotypes. Some evidence is presented is that, at least in some cases, a MMR-defective mechanism is responsible. However, unexpectedly, cultures starved for 100-day periods do not show these effects, and in many cases they reflect a decrease in the mutation rate.

Response: We greatly appreciate the reviewer's positive comments here. Note that we consider the results of MMR- strains in M1 and S1 to be also consistent with what drift-barrier hypothesis predicts: when starting with high mutation rate, populations initially experience increased mutation loads, which enhances the advantage of antimutators; hence, MMR- populations are expected to evolve reduced mutation rates.

Overall, this work is valuable and raises intriguing questions. However, it does not feel like a sufficiently strong new picture emerges from this work that would provide for a more detailed or comprehensive understanding of the forces that drive the mutation rate. At best, the current picture remains fragmentary.

Response: Owing to the enormous time and effort involved, despite the development of substantial theory, there is little empirical work on the evolution of the mutation rate. Our results demonstrate that such changes can unfold on remarkably rapid time scales, the direction of change depending on both the ecological and population-genetic environments. In particular, we show that the L10 populations (with

intermediate environmental cycles) evolved exceptionally high mutation rates compared to the L1 populations, providing empirical support for the idea that more fluctuating environments facilitate the evolution of high mutation rates (the view being that L100 lines experience almost a continuous environment of low resources). Thus, we believe that we provide novel insights into the understanding of mutation-rate evolution, and have substantially re-written the manuscript to make these arguments more coherent and comprehensive.

While the authors state that the wild-type data are consistent with the DBH, these effects are mostly very small. It may also be that the comparison to the ancestral rate is limited by the fact that there are only two ancestral values to compare to (one would have liked to see several more to allow a fair comparison between evolved and non-evolved rates).

Response: Although only two ancestral clones are used in the experiment, their mutation rates are similar to each other and to the mutation rates obtained in the previous MA/WGS literature (ref. 23). Therefore, our observation is unlikely to be an artifact of the sample size in our experimental design.

In contrast to the wt data, the MMR- data do not conform to the DBH hypothesis and, in fact, move in the opposite direction. So, a consistent pattern on the DBH has not emerged.

Response: Although our MMR- data in L1, M1, and S1 show evolved reductions in mutation rates, which is opposite to the results of WT data (evolving higher mutation rates), this result is not inconsistent with the DBH. Rather, one interpretation of this observation is that the purifying selection against mutational loads (which is elevated in MMR- lines) is dominant in mutation-rate evolution, which is consistent with the drift-barrier hypothesis. To make this explanation more clear, we added sentences in the Results section (p6, lines 9-16).

Other observations that limit the reaching of overarching conclusions from this work is that the results for the 10-day starvation regime differ dramatically from those of the 100-day starvation. Ad hoc explanations for all of these diverse findings can be advanced, as the authors do, and they do not sound unreasonable, but all this does not help with formulating consistent models that might be helpful to readership.

Response: We designed the experiments of L10 and L100 to test the consistency of mutation-rate evolution under different ecological / demographic challenges. Thus, it is of interest that the results under 10-day cycles are unique compared to other tested ecological contexts and cannot be fully explained by the drift-barrier hypothesis, whereas they are consistent with a previous theoretical study (ref. 20). We have revised the part of manuscript on explaining the relationship between hypermutator accumulation and the length of replenishing cycles in the revised manuscript to help the readers appreciate this issue (p9-10). We do not regard the fact that biology sometimes cannot be reduced to one-dimensional interpretations as a weakness of our study; in the next point, the reviewer seems to agree.

Finally, even for the two parallel cultures that make up a given treatment group, the results are often quite dissimilar. For example, clones A and B for the MMR- populations (Fig. 1B) are very dissimilar, the same for clones C and D (Fig. 1D) for the wt populations. Even within one clone, several disparate results

are observed. For example, see A1 and A2 for MMR- strain (Fig. 1B) or D1 and D2 for the wt strain (Fig. 1D). All of this can best be summarized by saying that evolution in individual cultures can follow different paths and that similar outcomes are often not to be expected. This is, in principle, very interesting and intriguing, and constitutes impetus for all kinds of further explorations.

Response: We agree that evolution in individual cultures can follow different paths and that similar outcomes are often not expected. A high level of within-population variation can arise from the recurrent input of mutation-rate variants during clonal evolution, leading to substantial wandering (at least several fold) of the mutation rate over evolutionary time (i.e., there is drift around the drift barrier). More discussion on this point is added (p10, lines 6-17).

However, in my view, the current picture that emerges here is too fragmentary and does not provide a sufficiently coherent message that would be of interest to a wider readership.

Response: Environmental changes are frequent in the nature and thus are important forces driving the evolution of natural populations. Understanding variation of mutation rates and studying the underlying mechanisms enhances our knowledge on how organisms respond to different environmental settings. How much the genetic adaptation to different environments, especially environments with a high heterogeneity, impacts mutation-rate evolution remains substantially unknown. To address this issue, we have used experimentally evolved *Escherichia coli* populations as a model system to understand the key ecological effects of fluctuating resource availability and population-size shrinkage on the dynamics of mutation-rate evolution. In this large and extremely laborious study, we confirmed that mutation-rate evolution can be impacted by demography, consistent with the DBH. However, it is important to learn that this hypothesis cannot explain fully mutation-rate evolution under environments with more fluctuating settings, which shows a non-monotonic relationship with the lengths of replenishing cycles. To this end, we have described a model to explain how ecological factors can facilitate or hinder the accumulation of hypermutators or antimutators, particularly in changing environments, expanding current theoretical thought on mutation-rate diversity in a way that accommodates the reality of real organisms. We re-organized the Introduction and Discussion, added more related explanation, to make this point more clearly.

One other important point:

For the L10 and L100 experiments, no information is given about the health or sustainability of the cultures during and after the 10- or 100-day starvation periods. It is my personal experience that *E. coli* cultures after they reach stationary phase start to decline in viable titer if maintained at 37C within days. There is also a considerable amount of literature on this subject. To be honest, I would be surprised if the cultures as used here would not have suffered many logs of viability loss. This type of information should have been provided. Any killing would obviously have consequences for the experimental outcomes as they reduce the effective population size. This may well provide an explanation for the intriguing differential L10 and L100 results, as the effects on population size may be dramatically different for the two conditions. This size difference may allow expression of mutator alleles in one case but not the other.

Response: As indicated by the reviewer, long-term starvation is a harsh environment for *E. coli*. However, *E. coli* populations can survive long-term starvation for many years, as observed in many other

experimental evolution studies (for examples, PMID: 16415927 and 28567442). One recent experimental evolution study even reports five parallel cultures under long-term starvation and without adding any new resource in three years (PMID: 33734381).

In our case, whole-population extinction events only occurred in some L100 populations before the first or the second transfer, and backup cultures replaced them to continue the experiment. During the experimental evolution, L10 and L100 populations did show smaller population sizes before the transfer than L1 populations, but the difference between L10 and L100 is at most 10-fold; the difference between L10 and L1 is at most 100-fold (ref.22; Extended Data Figure 7). Given such a small difference, we conclude that other factors are needed to explain why the higher mutation rates evolved in L10 populations than in L100 populations.

Reviewer #3 (Remarks to the Author):

Review of “Rapid evolution of mutation rate and spectrum in response to environmental and population-genetic challenges” by Wei et al.

This interesting manuscript describes a simple and important experiment: testing the evolution of mutation rates in *E. coli* during 1000 days of evolution under different setups of selection, population size, and baseline mutation rate. The results are very interesting and open additional questions: mutation rates changed significantly during the experiment, either up (in most wildtype populations) or down (in most MisMatch Repair populations). Population size (the dilution factor) had a minor effect, but the effect of selection was dramatic and non-monotonous: L10 groups (transfer every 10 days) showed significantly elevated mutation rates in most groups, in comparison with both L1 and L100 groups (transfer every 1 and 100 generations, respectively).

I think the manuscript is important and interesting, but its presentation requires revision. A broader context of different theories for the evolution of mutation rates should be included, and the results need to be discussed in that context.

Response: We thank the reviewer for positive comments and suggestions.

Specific comments:

Introduction

p. 1, Lines 4-6 “Across the Tree of Life, mutation rates vary by three orders of magnitude, and are inversely correlated with the effective population sizes (N_e) of species 5,6. Thus far, this observation has been best explained by the drift-barrier hypothesis (DBH)”

A broader context should be included. Multiple factors are commonly suggested (body size, generation time, metabolic rate, environmental changes), and some of them have complex interactions with population size (e.g. large body size is correlated with small effective population size).

Response:

In the revised manuscript, we have added more sentences to discuss these possibilities in the Discussion section:

“Across the Tree of Life, mutation rates vary by three orders of magnitude and inversely correlate with the effective population sizes of species, and this observation has been best explained by the drift-barrier hypothesis thus far³⁶⁻³⁹. Although other explanatory factors, such as body size, metabolic rate, and generation time^{40,41} have been proposed, the focus in these studies has been primarily on animals, and all correlations are based on rates of molecular evolution rather than actual rates of mutation per cell division. Thus, even if these other factors show strong correlations with evolutionary rates, whether these traits can directly affect mutation-rate evolution is still questionable. For example, because organisms with larger effective population sizes tend to have small body sizes⁴², an inter-specific correlation between body size and mutation rate is also consistent with the drift-barrier hypothesis.”

Abstract and Discussion

Abstract : “These results ... are broadly consistent with the drift-barrier hypothesis for the evolution of mutation rates”. This claim requires further justification in the body of the manuscript.

p. 8 In 14 “our results are consistent with many of the expectations of the DBH”

Which expectations of the DBH are the results consistent with? Please elaborate. In fact, the naïve reader may consider the interesting results of the paper as examples for the role of selection and environmental changes in the evolution of mutation rates (compare L1, L10, L100), while the population size appears to play only a minor role here (comparison of L1, M1, S1). A discussion of what theories would best explain the results is missing, and I think it could contribute to the paper.

Response: Our results are consistent with two main expectations of the DBH. First, mutation-rate evolved to be higher in smaller WT populations (S1 and M1) and to be lower in MMR- populations, consistent with DBH prediction that the selection towards lower mutation rates is prominently influenced by population size differences as well as the magnitude of the ancestral mutation rate (i.e., distance from the drift barrier). Second, evolving populations are expected to harbor relatively high levels of within-population variation in the mutation rate as selection against mutator alleles only takes place after sufficient time has elapsed for the development of a linked load of deleterious mutations. We discuss this in more detail in the revised manuscript.

However, the DBH cannot fully explain mutation-rate evolution under environments with more fluctuating settings. For this reason, we introduced a more general model to better explain how ecological factors can facilitate or hinder the accumulation of hypermutators or antimutators (see Discussion).

p. 4 lines 18-21 “These outcomes may be a consequence of a reduction in the efficiency of selection on the mutation rate resulting from increased genetic drift under N_e reduction in WT populations, but an offsetting of this effect by selection in MMR- backgrounds that generate extremely high mutation loads” This is one possible explanation. Can the authors discuss other explanations as well? could selection, for example, be responsible for the increased mutation rate in WT populations of all sizes in a rapid growth setup?

Response: In theory, the increase of mutation rate can be due to the hitchhiking effects during the

adaptation as well. However, the selection for rapid growth cannot be the only explanation for these data, as the MMR- and WT lines both increased experienced increased fitness during experimental evolution (ref. 25), although only WT lines evolved higher mutation rates.

Updated sentence: “In WT populations, mutation-rate evolution is most likely to be driven by a reduction in the efficiency of selection on the mutation rate resulting from reduced N_e or strong hitchhiking effects for hypermutators, which is not the case in MMR- populations. In fact, both MMR- and WT populations evolved to be fitter during the experimental evolution²⁵, but only WT populations evolved to have higher mutation rates. Therefore, the effect of increasing mutation rates was likely overwhelmed by much stronger selection against high mutation loads in MMR- populations.”

P.4 ln 23 – p.5 lines 1-2 “Although a recent study claims to have falsified the DBH [23], that work was based on indirect inference, not on experimental evolutionary analysis, and moreover relies on erroneous interpretations of the predictions of DBH theory.”

If the authors want to discuss this disagreement, additional details of [23] and the relevant predictions of DBH are needed.

Response: More elaboration has been added (p10-11). An additional problem is that the cited study is now in question because of errors in construction and analysis of the experimental lines, and that an attempt to repeat the work by the original authors is now underway.

p. 5, lines 15-end of paragraph

“Why is there a particularly high mutation rate in L10 populations but not in L100 populations?...”

This is a very puzzling question, and a deeper investigation of it may contribute to the paper. The explanation suggested by the authors involves “fluctuating environment with intermediate cycle length”. However, several parameters are different between L10 and L100: the stress is more dramatic in L100 (mentioned by the authors – but by how much? Can the authors evaluate? and may the stress affect mutation rates?), the number of generations is longer in L10 (by how much?), and of course the period between changes is indeed longer in L100. How are these factors combined? A model could perhaps help here, or at least a deeper discussion of the different factors and their interaction.

Response: We introduced a model to better explain how ecological factors can facilitate or hinder the accumulation of hypermutators or antimutators (see Discussion), resulting in a non-monotonic relationship with the lengths of replenishing cycles.

Figure 2

The caption of figure 2 is very short and minimalistic, and the text does not refer to Fig. 2 in detail (only in p. 3 line 3 “(Figs. 1B-E, 2; Extended data Table S1; Extended data Fig. 4 S2)”). Which of the results detailed in the text are presented in Fig. 2? Please also compare Fig. 1B-E and Fig. 2.

Response: As figure 2 provides redundant information from figure 1, we deleted figure 2. Fig. 1B-E have become the new figure 2 in the revised manuscript.

Reviewers' Comments:

Reviewer #1:

Remarks to the Author:

The authors addressed all my concerns with additional experiments and data analyses. The interpretation of the results in the revised manuscript is careful. I have no further comments.

Reviewer #2:

Remarks to the Author:

The authors have addressed all comments raised by the reviewers. As a result, the manuscript is significantly improved.

Reviewer #3:

None